# Revitalizing Cancer Treatment: Exploring the Role of Drug Repurposing

**DOI:** 10.3390/cancers16081463

**Published:** 2024-04-11

**Authors:** RamaRao Malla, Sathiyapriya Viswanathan, Sree Makena, Shruti Kapoor, Deepak Verma, Alluri Ashok Raju, Manikantha Dunna, Nethaji Muniraj

**Affiliations:** 1Cancer Biology Laboratory, Department of Biochemistry and Bioinformatics, GITAM School of Science, GITAM (Deemed to be University), Visakhapatnam 530045, Andhra Pradesh, India; 2Department of Biochemistry, ACS Medical College and Hospital, Chennai 600007, Tamil Nadu, India; vvspriya@gmail.com; 3Maharajah’s Institute of Medical Sciences and Hospital, Vizianagaram 535217, Andhra Pradesh, India; 4Department of Genetics, University of Alabama, Birmingham, AL 35233, USA; 5Department of Oncology, Johns Hopkins University School of Medicine, Baltimore, MD 21231, USA; 6Medicover Hospitals, Visakhapatnam 530022, Andhra Pradesh, India; 7Center for Biotechnology, Jawaharlal Nehru Technological University, Hyderabad 500085, Telangana, India; 8Center for Cancer and Immunology Research, Children’s National Hospital, 111, Michigan Ave NW, Washington, DC 20010, USA

**Keywords:** cancer, drug resistance, drug repurposing, computational approaches, drug repositioning combination therapies, protein re-engineering, nanotechnology, therapeutics

## Abstract

**Simple Summary:**

Drug repurposing is emerging as a promising avenue for addressing clinical challenges in treating drug-resistant and metastatic cancers. As a case study, some FDA-approved drugs and re-engineered proteins have been revitalized to target various mechanisms in sensitizing drug-resistant cancers. Various computational approaches, including genome-wide association studies, machine learning, artificial intelligence, and deep learning, have been discussed for repurposing drugs for cancer treatment. The drug repurposing approach holds great promise in revolutionizing cancer therapeutics.

**Abstract:**

Cancer persists as a global challenge necessitating continual innovation in treatment strategies. Despite significant advancements in comprehending the disease, cancer remains a leading cause of mortality worldwide, exerting substantial economic burdens on healthcare systems and societies. The emergence of drug resistance further complicates therapeutic efficacy, underscoring the urgent need for alternative approaches. Drug repurposing, characterized by the utilization of existing drugs for novel clinical applications, emerges as a promising avenue for addressing these challenges. Repurposed drugs, comprising FDA-approved (in other disease indications), generic, off-patent, and failed medications, offer distinct advantages including established safety profiles, cost-effectiveness, and expedited development timelines compared to novel drug discovery processes. Various methodologies, such as knowledge-based analyses, drug-centric strategies, and computational approaches, play pivotal roles in identifying potential candidates for repurposing. However, despite the promise of repurposed drugs, drug repositioning confronts formidable obstacles. Patenting issues, financial constraints associated with conducting extensive clinical trials, and the necessity for combination therapies to overcome the limitations of monotherapy pose significant challenges. This review provides an in-depth exploration of drug repurposing, covering a diverse array of approaches including experimental, re-engineering protein, nanotechnology, and computational methods. Each of these avenues presents distinct opportunities and obstacles in the pursuit of identifying novel clinical uses for established drugs. By examining the multifaceted landscape of drug repurposing, this review aims to offer comprehensive insights into its potential to transform cancer therapeutics.

## 1. Introduction

Cancer, often referred to as the ‘pathology of the century’, remains a formidable global challenge. Significant strides have been made in understanding the disease and refining its diagnosis, primarily due to advanced techniques like single-cell analysis. RNA sequencing, at the molecular level, has facilitated a comprehensive understanding of various aspects of cancer, including tumor cell heterogeneity and the mechanisms underlying drug resistance. Bioinformatics has been instrumental in identifying diverse treatment strategies, targeting specific molecular pathways. The establishment of The Cancer Genome Atlas has significantly enriched our comprehension of this field [1]. However, despite these advancements, cancer remains the leading cause of mortality and exerts a substantial economic burden on society [2]. 

The complexity of cancer is further compounded by the resistance of tumor cells to current medications [3,4]. This resistance primarily stems from uncontrolled metastasis, which accelerates disease progression and reduces treatment efficacy through mechanisms such as inadequate vascularization, hypoxia, increased intertumoral pressure, and drug-induced phenotypic resistance [5,6]. Consequently, there is an urgent need to identify alternative drugs capable of counteracting drug resistance and impeding disease spread [7]. The development of new drugs is not only costly but also time-consuming, often taking more than a decade to transition to clinical practice and assess their efficacy as the preferred treatment option. Repurposed drugs are estimated to take about 6.5 years and cost around USD 300 million to bring to market, compared to 13 to 15 years and USD 2–3 billion for novel drugs. Moreover, repurposed drugs are less likely to fail overall in clinical trials [8,9].

Drug repurposing encompasses the repositioning of off-patent FDA-approved drugs, failed drugs, and patented drugs for new clinical indications. Generic drugs, off-patent and widely available, are favored for repurposing due to their extensive safety and efficacy data, lower cost, and reduced risk [9]. Patented drugs, protected by patents and typically in late-stage trials, have limited accessibility to safety and efficacy data. Failed drugs, which did not receive approval due to various reasons, present an opportunity for repurposing, especially for companies aiming to salvage investments. However, accessing safety and efficacy data for failed drugs is challenging, and addressing initial failure issues is crucial for their successful repurposing [9]. One significant advantage in repurposing existing drugs is the wealth of comprehensive data available, including pharmacodynamics, pharmacokinetics, toxicity, and safety profiles in humans. This extensive knowledge base provides a valuable head start in the drug development process [10,11]. Additionally, drug repurposing offers a rapid, cost-effective, and potentially less toxic alternative to traditional anticancer drugs, often bypassing preclinical and phase I trials and directly entering phase II clinical trials [8].

Various comprehensive drug-repurposing methods, including knowledge-based, drug-based, activity-based, in silico, and in vitro approaches, are employed to identify effective drugs for specific conditions [12]. Experimental strategies for drug discovery are categorized as target-based or drug-based, with examples such as tamoxifen, which transitioned from a failed contraceptive pill to an FDA-approved breast cancer treatment [13]. In silico methods boast advantages such as robust algorithms for gene expression analysis, cost-effectiveness, time efficiency, and easy access to available sources [14]. Increasingly, computational methods like signature matching screens and genome-wide association studies (GWAS) are favored. For instance, cimetidine, an anti-peptic ulcer drug, has shown promise for lung adenocarcinoma, while anti-psychotic imipramine has exhibited potential for glioblastoma [9]. Artificial intelligence (AI), machine learning (ML), and deep learning aid (DL) in uncovering associations between drugs and diseases through text mining studies [15], such as repurposing aspirin for targeting TP53 and metformin for cancer [9]. Furthermore, chemical similarity analysis and molecular docking simulations assist in identifying repurposing candidates, exemplified by thalidomide’s potential for colon and renal cancer [9].

While drug repositioning holds promise, it faces notable limitations. Firstly, patenting issues may obstruct the process due to existing intellectual property rights restricting the repositioning of certain drugs. Secondly, financial constraints, including the costs of obtaining licenses and conducting extensive clinical trials, can impede progress. Additionally, the efficacy of repurposed drugs as monotherapy may fall short, necessitating the development of combination therapies. These challenges highlight the complexity of drug repositioning and stress the importance of careful planning. Moreover, some drugs may require higher doses or prolonged administration, prompting comprehensive investigations into potential side effects, which could compromise repositioning efforts [16,17,18]. Despite these constraints, addressing them is vital to fully utilize drug repositioning’s potential. This review offers an extensive examination of the significant role played by artificial intelligence, machine learning, and deep learning in drug repurposing, alongside various drug delivery systems, including self-delivery drugs. These innovative approaches show promise in advancing cancer treatment and enhancing patient outcomes.

## 2. Experimental Approach: Repurposing Some Anti-Alcoholic, Anthelmintic and Antiepileptic Drugs for Cancer, Using Breast Cancer as a Case Study

Breast cancer (BC) stands as a leading cause of mortality in women globally, primarily attributed to its high metastatic ability, resistance to therapy, and the tendency for recurrence [19]. The inherent heterogeneity of BC manifests in distinct subtypes, each characterized by unique molecular drivers and clonal expansion capacities. While a broad spectrum of targeted therapeutics has been developed and is employed routinely for the treatment of specific BC subtypes, challenges persist, particularly in addressing the triple-negative subtype, for which treatment options remain limited. Furthermore, therapy resistance remains a formidable challenge in comprehensively treating all BC subtypes [20,21]. The process of discovering novel drugs to combat breast cancer, aiming to replace existing medications, is both time-consuming and costly. To overcome the challenges and limitations associated with anticancer drugs, repurposing clinically validated drugs offers several advantages, including minimal side effects, limited development of resistance, and reduced cost [22]. In recent years, repurposed agents have been increasingly demonstrated as adjuvants for cancer treatment, encompassing specific anti-alcoholic, anthelmintic, and antiepileptic agents [23]. They sensitize breast cancer (BC) by targeting different mechanisms (Figure 1).

### 2.1. Sensitization of Drug-Resistant BC Cells by Anti-Alcoholic Drugs

Anti-alcoholic drugs are the primary targets of aldehyde dehydrogenase. Disulfiram (DSF), originally an anti-alcohol drug clinically used for the treatment of alcohol addiction, functions as a copper-binding protein, serving as a copper ionophore [24]. This property enables DSF to trigger copper-mediated oxidative stress by generating reactive oxygen species (ROS) [25]. Recently, DSF has been repurposed for solid tumor treatment as an adjuvant, including BC [26]. It inhibits BC growth by targeting various cellular and signaling mechanisms (Figure 2a). Askgaard et al. have presented evidence supporting the protective effect of DSF against BC in epidemiological studies [27]. DSF reduced TGF-mediated EMT in MCF-7 and MDA-MB-231 cells by suppressing the activity of NF-kB. Additionally, DSF decreased stem-like characteristics, invasion, and migration of BC cells by blocking the ERK/Snail signaling pathway [28]. The treatment of MDA-MB-231 cells with a DFS and PI3K inhibitor (LY294002) inhibited cell proliferation by targeting Akt signaling [29]. The combination also inhibited the growth of tumors in xenograft mice implanted with MDA-MB-231 cells. Sweetha et al. demonstrated that DSF enhances the sensitivity of MCF-7 and 4T1 BC cells to docetaxel, inducing cytotoxicity, apoptosis, and autophagy through ROS generation [30]. It also induced cytotoxicity in MCF-7 and BT474 cells by enhancing endo-lysosomal zinc levels. It spatially instigated the lysosomal and late endosomal disorganization [31]. Another study revealed that triple treatment with DSF, doxorubicin (DOX), and hydrazine (Hyd) sensitizes both DOX-resistant MCF-7 cells and their wild-type counterparts by inducing apoptosis [32]. Remarkably, DSF decreased the required dose of DOX to induce cytotoxicity in DOX-resistant cells.

Cancer stem cells (CSCs) are a small population of cells within a tumor characterized by their unique ability to self-renew through division and generate diverse cell types comprising the tumor, which are often associated with drug resistance [33,34]. The combination of DSF/Cu sensitizes BCSCs through various mechanisms (Figure 2b). The DSF/Cu combination enhanced paclitaxel (PTX)-induced cytotoxicity and reduced clonogenicity in breast CSCs. It induced apoptosis by upregulating BAX expression and downregulating Bcl-2 through activating ROS-dependent JNK/MAPK pathways. Moreover, the combination reduced mammosphere formation by targeting CD44^high^ and CD24^low^ CSCs, diminishing ALDH1 expression by inhibiting the NF-kB pathway [35]. Adding 6-thioguanine (6-TG) to DSF and Cu complex synergistically induces DNA damage in MDA-MB-231 cells. This combination leads to cell cycle blockage and promotes apoptosis. These effects are associated with a reduction in total and phosphorylated ATR (ataxia telangiectasia and Rad3-related) proteins and checkpoint kinases [36]. DSF has been investigated for its potential to target CSCs. Yang et al. demonstrated that DSF sensitizes cisplatin (CIS)-resistant aldehyde dehydrogenase (ALDH)-positive CSCs isolated from MCF-7 cells by triggering the generation of ROS as well as reducing the expression of transcription factors related to CSCsm such as Sox, Oct, and Nanog [37]. Moreover, Kim et al. demonstrated that combining DSF with Cu suppressed CD44^high^ and CD24^low^ CSCs isolated from MDA-MB-231 cells, inducing apoptosis through increased caspase 3 activity. Mechanistically, the DSF and Cu combination reduced the survival of BCSCs by targeting STAT3 signaling, evidenced by decreased phosphorylated STAT3 and downstream mediators survivin and cyclin D1. Additionally, it attenuated stem-like characteristics by repressing ALDH1A1 and CD44 levels in a xenograft mouse model bearing MDA-MB-231 cells [38]. DSF/Cu induced apoptosis in CSCs isolated from HER2-positive BC cells by blocking HER2/Akt signaling. Additionally, it diminished mammosphere formation by suppressing the expression of ALDH1 [39]. DSF-encapsulated NPs, comprising poly(lactide-co-glycolide) (PLGA) and polyethylene glycol (PEG), selectively targeted the folate receptor in BC cells. The NPs facilitated the targeted delivery of DSF, resulting in the inhibition of cell proliferation and induction of apoptosis through the generation of ROS. Moreover, these NPs effectively reduced established tumor growth in a mouse model [40]. DSF-encapsulated liposomes sensitize pan-resistant CSCs isolated from BC cells. This sensitization is mediated by targeting mammosphere formation by inhibiting the NF-kB pathway [41].

Liu et al. demonstrated that DSF targets the tumor microenvironment (TME) by inhibiting the growth of BCSCs through the reduction of ALDH1A1 function. Additionally, it triggered antitumor activity by inhibiting the expansion of myeloid-derived suppressor cells, which is mediated by reducing GM-CSF secretion [42]. A dual enzyme/pH-responsive nanoparticle (NP) loaded with DSF and DOX effectively induced cytotoxicity in 4T1 BC cells and inhibited wound healing and metastasis [43]. Also, magnetic NPs conjugated with DSF enhanced cytotoxicity in MCF-7 cells in combination with sodium nitroprusside and Cu [44]. A different study demonstrated that liposomes encapsulated with DOX and DSF induced cytotoxicity in p-glycoprotein (Pgp)-overexpressing BC cells by controlled release of DOX and DSF. The released DSF promotes the ubiquitination of Pgp via inducing sulfhydration [45].

In a recent study, Chu et al. proved that a complex of DSF and Cu effectively inhibited the growth of MDA-MB-231 cells through the induction of ferroptosis. Mechanistically, the complex elevated intracellular iron levels triggered ROS release, leading to malondialdehyde formation and the induction of mitochondrial atrophy. The ferroptotic signaling activated this ferroptosis process, upregulating its target genes associated with redox homeostasis [46]. Kim et al. demonstrated that the complex of Cu and DSF effectively reduced the invasion and migration of MDA-MB-231 cells by decreasing focal adhesions, promoting vimentin degradation, and activating calpain. Additionally, in a xenograft model, the complex exhibited a reduction in the growth of established tumors and their localization in the lungs through calpain activation [47]. The nanofiber, composed of DSF, cellulose acetate (CA), and poly (ethylene oxide) (PEO), exhibits safety towards normal cells while demonstrating potential cytotoxicity towards MDA-MB-231 cells. It induces apoptosis, inhibits ALDH1A1, and hinders BCSCs [48]. Wang et al. demonstrated that DSF, in combination with copper and ionizing radiation, reprogrammed chimeric antigen receptor T (CAR-T) cells, enhancing effective cytotoxicity and expansion and reducing exhaustion. The reprogrammed CAR-T cells exhibited a capacity to diminish the growth of established tumors in a xenograft mouse model. Furthermore, the combination of DSF and CAR-T cell treatment attenuated metastatic features by converting the cold TME into a hot tumor [49].

Naltrexone (NTX), an antagonist of opioid receptors known for its use in treating alcohol dependence, has been investigated for its potential anticancer properties. Low doses of naltrexone (LDN) have hindered the proliferation of cancer cells, diminished tumor growth by targeting signaling pathways, and modulated the immune system [50]. A recent clinical study assessing the impact of LDN on ER-positive BC patients with metastasis showed that LDN exhibited high tolerability and demonstrated modest activity against ER-positive BC [51]. Murugan et al. demonstrated that the combination of naltrexone (NTX) and propranolol, a β-2 adrenergic receptor antagonist, effectively reduced the progression of BC growth. This reduction leads to a decrease in the expression of proteins associated with EMT, a lowered release of inflammatory cytokines, and an enhancement in the expression of apoptotic proteins. The combined treatment exhibited notable antitumor activity in a preclinical BC model [52]. These studies suggest that the Antabuse drugs disulfiram and naltrexone can sensitize drug-resistant BCs.

### 2.2. Sensitization of BC Cells with Anthelmintic Drugs

Albendazole (ABZ), mebendazole (MBZ), and flubendazole are benzimidazole-based anthelmintic drugs known to induce cell death in parasites and mammalian cells. This mechanism inhibits glucose uptake and transport, which targets microtubule systems [53]. They have been repurposed for anticancer activity in clinical studies. ABZ and MBZ induced apoptosis in HT29 colon cancer cells and MCF-7 and MDA-MB 231 BC cells by activating caspase 3 activity. The treatment displayed apoptotic characteristics in both types of cancer cells, including DNA fragmentation, increased permeabilization of the mitochondrial membrane, and translocation of phosphatidylserine to the outer layer of the plasma membrane. Additionally, they blocked the cell cycle at the G2M phase by generating ROS and disrupting tubulin polymerization [54]. Jubie et al. reported that analogs of ABZ and MBZ exhibited potent cytotoxicity comparable to Lapatinib in BC cells overexpressing HER2 [55]. Furthermore, polyurethane encapsulation augmented the cytotoxicity of ABZ in MCF-7 and MDA-MB 231 cells by inducing alterations in cell morphology and fragmentation of nuclear DNA [56]. ABZ treatment reduced the proliferation as well as migration of MDA-MB-231 cells. The treatment also induced apoptosis by targeting the GLUT1/AMPK/P53 signaling axis. In the xenograft mouse model of MDA-MB 231, the treatment reduced the tumor volume and standardized uptake value with increased apoptotic cells in the treated group [57]. Priotti et al. demonstrated that the inclusion complex of cyclodextrin and ABZ reduced the growth of 4T1 cells and inhibited tumor growth in BALB/c mice [58].

MBZ has been established as clinically safe with low toxicity and has demonstrated efficacy in preclinical studies against various cancers. In the context of TNBC, MBZ has been shown to decrease cell proliferation by arresting the cell cycle at the G2M phase and inducing apoptosis. Additionally, it has proven effective in reducing tumor growth and lung and liver metastasis. Furthermore, MBZ can diminish CSC characteristics, attributed to its ability to target the expression of β4 integrin [59]. Additionally, MBZ reduced the hypoxia response in BC cells and a mouse model of BC by targeting both HIF-1α- and HIF-1β-mediated transcription [60]. Chitosan NPs loaded with MBZ and targeted with folic acid (FA) were shown to reduce tumor volume in a BALB/c mice model of 4T1 TNBC cells. This formulation significantly decreased the number of metastatic colonies in the liver, leading to an improved survival rate of approximately 50% [61]. Another study reported that micelles containing mebendazole (MBZ) reduced the expression of vascular endothelial growth factor (VEGF), akin to the established kinase inhibitor sorafenib [62].

MBZ effectively reduced the CSC population and mammosphere formation ability induced by radiation therapy in TNBC cells. MBZ induced cell cycle arrest at the G2M phase by triggering double-stranded breaks in DNA, as evidenced by increased γ-H2AX foci, and induced apoptosis. Moreover, it demonstrated the ability to control tumor growth by sensitizing TNBC cells in a xenograft tumor model [63]. It also sensitized radiotherapy-resistant (RT-R) MDA-MB-231 cells by triggering DNA damage, halting the cell cycle by reducing the expression of ESM, a cancer progression-related protein. Also, it reduced the stemness and expression of stemness-related proteins CD44 and Oct3/4 [64]. Additionally, MBZ sensitized radiotherapy-resistant (RT-R) MDA-MB-231 cells by inducing cytotoxicity via triggering DNA damage and stimulating natural killer (NK) cell-dependent cytotoxicity [65].

Flubendazole (FLBZ), a clinically proven anthelmintic drug, has been repurposed as a novel anticancer agent. It is recognized for inducing autophagy and apoptosis in TNBC [66]. Mechanistically, it induced autophagy in MDA-MB 231 cells by upregulating the expression of EVA1A, a key regulator of programmed cell death [67], and downregulating autophagy-related protein 4B(ATG4B) as well as inducing release of ROS [68]. Moreover, it diminished CSCs by inhibiting mammosphere formation and suppressing the expression of SOX2, Oct4, and Nanog. Additionally, FLBZ lowered the expression of N-cadherin and Vimentin while increasing Keratin, indicating its ability to induce the differentiation of CSCs. Furthermore, it induced G2M cell cycle arrest by promoting the formation of a monopolar spindle through the blockade of tubulin polymerization. Notably, FLBZ sensitized BCSCs to anticancer drugs DOX and 5-fluorouracil (5-FU) by enhancing cytotoxic activity [69]. In HER-2 positive BC, it suppresses BCSCs by inducing apoptosis by targeting HER/Akt signaling. Additionally, it demonstrates inhibition of mammosphere formation and the stemness of the CD44^high^ and CD24^low^ cell populations by targeting the expression of ALDH1. Furthermore, by downregulating CSC markers, FLBZ-sensitized trastuzumab-resistant TNBC in a xenograft model [70]. Furthermore, FLBZ demonstrated reduced metastasis and the stemness of BCSCs derived from TNBC cells. It effectively inhibited mammosphere formation and decreased the CD44^high^ and CD24^low^ side populations by targeting ALDH1 in BCSCs. Additionally, FLBZ exhibited the capability to reduce lung and liver metastasis by targeting angiogenesis by inhibiting STAT3 signaling and downregulating MMP-2 and MMP-9 expression [71]. FLBZ was demonstrated to sensitize BC cells resistant to PTX by targeting the HIFα-dependent PI3K/Akt pathway [72]. These studies recommend the repurposing of anthelmintic drugs, specifically ABZ, MBZ, and FLBZ, for clinical evaluation against breast cancer subtypes, with a particular emphasis on TNBC.

### 2.3. Sensitization of BC Cells with Antiepileptic Drugs

Valproic acid (VPA) is a well-established inhibitor of histone deacetylase (HDAC) [73]. Due to its HDAC-inhibiting activity and its safe use as a chronic therapy for epileptic disorders [74], VPA has been considered a good candidate for anticancer therapy [75]. It has been experimentally validated for its efficacy in treating specific BC subtypes [76]. It inhibited the proliferation and colony formation ability of MDA-MB-231 and MCF-7 cells by suppressing the Warburg effect by reducing the M2 isoform of pyruvate kinase expression (PKM2). Mechanistically, VPA regulates PKM2 expression by inactivating ERK1/2-dependent HDAC1 [77]. Giordano et al. demonstrated that VPA inhibits the proliferation of MDA-MB 231 and MCF-7 cells by arresting the cell cycle at the G0/G1 phase through the generation of ROS from mitochondria. Furthermore, VPA induces apoptosis by releasing cytochrome c and activating poly (ADP-ribose) polymerase (PARP), achieved through the downregulation of Bcl2 and the upregulation of Bax and Bad [78]. In a separate study, Injinari et al. experimentally demonstrated that VPA induces apoptosis in MDA-MB-231 and MCF-7 cells by reducing the expression of HDAC by increasing the expression of miR-34a and 520 h. Additionally, VPA elicited cytotoxicity in these cells by enhancing lipid peroxidation [79]. It combines Berberine with photodynamic therapy (PDT) followed by VPA treatment, synergistically reducing viability in MDA-MB-231 cells [80]. This combination strategy effectively blocked colonization and enhanced apoptosis by inducing morphological alterations.

The expression of ALDHs has been associated with drug resistance in BC. VPA counteracted CIS-induced resistance in MDA-MB-231 cells. VPA treatment significantly altered metabolites, with acylcarnitines (ACs) and phosphatidylcholines (PCs) most affected in CIS-resistant MDA-MB-231 cells. Furthermore, the combination of VPA and CIS enhanced levels of PCs, glucose, and sphingomyelin. This combination sensitized CIS-resistant TNBC cells by reprogramming the metabolism of fatty acids and hexoses [81]. VPA treatment reduced the extracellular acidification rate in both CIS-resistant and sensitive TNBC cell lines, accompanied by reduced oxygen consumption and alterations in the cell cycle. VPA reprogrammed CIS-resistant TNBC cells by modulating ALDH expression, leading to altered glucose metabolism. Additionally, this study demonstrated that combining VPA with CIS and DSF effectively induced cytotoxicity in organoids derived from TNBC [82]. VPA enhanced the cytotoxic effect of MTX on the receptor-positive cell line MCF-7 but not on the receptor-negative BC cell line MDA-MB-231 [83]. This study suggests that the combination of MTX and VPA might exhibit receptor-specific actions in BC.

VPA sensitized HER2-overexpressing cells to trastuzumab by stimulating antibody-dependent cell-mediated phagocytosis by upregulating antibody-binding Fc-gamma receptors on monocytes. Concurrently, VPA downregulated myeloid leukemia cell differentiation protein with anti-apoptotic activity in BC cells. Furthermore, VPA induced immunogenic cell death in BC cells by eliciting calreticulin exposure and reducing CD47 expression on the cell surface, thereby altering the ‘do not eat me’ signal [84]. Su et al. demonstrated that VPA reduced the tumor burden by reprogramming the tumor microenvironment (TME) and promoting the recruitment of tumor-suppressing macrophages (M1) to the breast tumor site [85]. Furthermore, in a rat model of BC, VPA reduced tumor growth in both irradiated and non-irradiated tumor sites. Notably, VPA enhanced the recruitment of M1 macrophages to the non-irradiated tumor site by increasing the expression of IL-12 while decreasing TGF-β expression. Additionally, VPA induced cytotoxicity at the non-irradiated tumor site by promoting the infiltration of CD8+ T-cells via the release of Granzyme B. However, the combination of RT and VPA promoted an immunosuppressive TME by facilitating the polarization of tumor-associated macrophages to M1 macrophages [86]. These findings suggest that VPA modulates the TME by serving as an adjuvant in combination with radiotherapy or immunotherapy.

## 3. Re-Engineering Proteins for Drug Repurposing Using Nemvaleukin Alfa as a Case Study

Re-engineering proteins for drug repurposing is a promising strategy to repurpose drug development by leveraging existing knowledge about the structure, adverse effects, pharmacokinetics, and other drug properties [87]. For example, IL-2, a cytokine initially discovered in human peripheral blood leukocytes, demonstrated significant anti-tumor potential in the 1980s, notably in metastatic melanoma and renal cancer. In the late 1990s, a study spanning more than a decade showed promising responses in metastatic melanoma and renal cancers, with some patients maintaining a complete response for several years. The FDA approved high-dose IL-2 therapy for metastatic renal cell carcinoma and metastatic melanoma; however, other tumor types showed limited responses [88].

Recombinant IL-2, while effective in cancer treatment, poses challenges due to its short half-life and severe toxicities, including capillary leak syndrome and pulmonary edema. To address these limitations, novel therapies like nemvaleukin alfa are being developed. Nemvaleukin alfa, an IL-2-IL-2Rα fusion protein engineered a selective binding affinity for intermediate-affinity IL-2 receptor complexes. This selectivity enables the preferential activation of CD8-positive cytotoxic T cells and NK cells while concurrently suppressing the expansion of immunosuppressive Tregs (Figure 3). Unlike conventional recombinant IL-2, nemvaleukin alfa retains stability in circulation and does not necessitate metabolic activation. By specifically targeting receptor complexes, nemvaleukin alfa endeavors to alleviate the severe immune-related toxicities associated with high-dose IL-2 therapy [89]. The ARTISTRY-1 trial (NCT02799095) investigated the efficacy of nemvaleukin alfa in treating advanced solid tumors through various approaches, including dose escalation, monotherapy expansion, and combination therapy with pembrolizumab. Monotherapy exhibited promising anti-tumor activity, particularly in cutaneous melanoma, mucosal melanoma, and renal cell carcinoma, prompting FDA designations in 2021. Noteworthy partial responses were also noted in patients with platinum-resistant ovarian cancer, alongside instances of stable disease across various other cancers. These initial findings suggest that nemvaleukin alfa may offer advantages over other IL-2 variants [89,90].

## 4. Repurposing Drugs with Nanoliposomes Using Pancreatic Cancer as a Case Study

Pancreatic cancer (PC) is a leading cause of cancer-related mortality globally. The disease’s grim 5-year survival rate of only 9% primarily stems from late-stage diagnoses, where symptoms seldom manifest until the disease reaches an advanced, unresectable stage. Despite advancements in therapeutic options, particularly in comparison to other cancers, the response to standard chemotherapy remains unsatisfactory, with treatments like gemcitabine offering limited survival benefits and often exhibiting toxicity [91,92]. While certain targeted therapies show promise for specific genetic aberrations, such as BRCA1/BRCA2 mutations or NTRK1-3 fusion genes, they benefit only a small subset of patients. Similarly, immune checkpoint inhibitors (ICIs) hold potential for PC patients with defective mismatch repair (MMR), but their efficacy remains modest. Radical surgery offers the possibility of cure, yet only a minority of patients qualify, and relapse rates post-surgery are high, indicating an unmet need for PC [91,92].

Liposomal carriers are utilized to deliver anticancer medications directly to tumors. This delivery mechanism takes advantage of the immature and leaky blood vessels in tumors, as well as impaired lymphatic drainage at the tumor site. By targeting drug delivery to tumors, liposomes can potentially reduce systemic drug exposure while enhancing tumor exposure, thereby improving safety. Examples of this approach include nanoliposomal irinotecan (nal-IRI) [93,94]. nal-IRI comprises pegylated liposomal particles with a diameter of 111 nm, containing an irinotecan sucrosofate salt payload. The drug-to-phospholipid ratio is 473 mg irinotecan-HCl per mmol of phospholipid. The liposome’s phospholipid composition includes distearoylphosphatidylcholine, cholesterol, and pegylated 1,2-distearoyl-sn-glycero-3-phosphorylethanolamine in a molar ratio of 3:2:0 [93]. naI-IRI works by inhibiting the TOP1 enzyme, primarily through its active metabolite SN-38, which stabilizes the TOP1/DNA complex, causing DNA strand breaks, inhibiting cell replication, and leading to cell death. SN-38 exhibits significantly higher TOP1 inhibitory activity compared to irinotecan, up to 1000-fold greater (Figure 4). Irinotecan and its metabolites are eliminated from the body through a hepatobiliary pathway, being excreted in feces and urine by ABC transporters. The inactive metabolite SN-38G can be reactivated to SN-38 by β-glucuronidases in the human colorectum. Elevated levels of tumor β-glucuronidases may increase tumor exposure to SN-38 in vivo [93].

In the phase 3 NAPOLI-1 trial involving patients with metastatic PC who progressed on gemcitabine-based treatment, nal-IRI plus leucovorin (LV) and 5-fluorouracil (5-FU) led to improved progression-free survival (PFS) and overall survival (OS) compared to 5FU-LV alone. The FDA approved nal-IRI in 2015, recommending its use as a second-line treatment option for patients who have progressed on gemcitabine-based treatment [95,96]. Recent results from the NAPOLI-3 trial have demonstrated the advantage of combining nal-IRI with 5FU/LV and oxaliplatin (NALIRIFOX) over gemcitabine-nab-paclitaxel in the first-line treatment setting. This combination showed an improved OS of 11.1 months versus 9.2 months (HR 0.84; *p* = 0.04), as well as an enhanced PFS of 7.4 months compared to 5.6 months (HR 0.70; *p* = 0.0001) [97]. On 13 February 2024, the FDA granted approval for naI-IRI in combination with oxaliplatin, 5FU/LV, for the first-line treatment of metastatic PC [98].

After over ten years, the FDA granted approval for a novel first-line treatment for individuals diagnosed with metastatic PC [98]. The success of nal-IRI highlights the potential of nano-scale delivery methods, suggesting that compounds with high toxicity profiles could be effectively targeted to tumor environments. This targeted delivery approach may reduce the occurrence of adverse off-target effects while enhancing treatment efficacy.

## 5. Computational Approaches for Cancer Drug Repurposing

### 5.1. Genome-Wide Association Studies (GWAS)

GWAS can play a pivotal role in drug repurposing efforts, as they uncover crucial biological insights into complex traits that can aid in identifying compounds suitable for repurposing. GWAS have identified numerous variants associated with clinically relevant phenotypes, shedding light on genes, pathways, and genetic overlap between different traits. While GWAS traditionally focus on common variants due to their greater power and ease of imputation, the combined effect of many common variants significantly contributes to overall trait heritability. The statistical genetics community has developed sophisticated methods to integrate GWAS with other ‘omics’ data sets, enabling the identification of drug repurposing opportunities [99]. On such advance is the study done by Cheng et al., where they leverage advancements in DNA/RNA sequencing to rapidly identify novel therapeutic targets and repurpose existing drugs for various diseases by precisely targeting individualized disease modules [100]. Cheng et al. present a novel algorithm called the genome-wide positioning systems network (GPSnet) for repurposing drugs. Their approach involves focusing on disease modules identified from individual patients’ sequencing data mapped to the human protein–protein interactome network. By analyzing sequencing data from a substantial cohort of approximately 5000 patients spanning 15 cancer types sourced from The Cancer Genome Atlas, they demonstrate the efficacy of GPSnet in accurately predicting drug responses and identifying potential new uses for 140 approved drugs. Particularly noteworthy is the experimental validation of ouabain, a medication approved for treating cardiac arrhythmia and heart failure, which exhibits promising anti-tumor effects in lung adenocarcinoma by targeting a HIF1α/LEO1-mediated cell metabolism pathway [100].

### 5.2. Structure-Based Repurposing

Molecular docking is a structure-based drug design tool, which enables efficient utilization of computational resources and swift evaluation of millions of compounds [101]. Various commercial molecular docking tools, such as CovDock, FLEXX, GOLD, ICM-Pro, DOCKTITE, Molecular Operating Environment (MOE), MacDOCK, CovalentDock, and AutoDock 4, are available for drug discovery and drug repurposing [102].

Various studies have shown docking can be used for drug repurposing. For example, Shaikh et al. conducted docking simulations of 112 chemotherapeutic drugs against 18 validated kinase targets across nine cancer types [101]. Various molecular modeling tools such as MOE, Cresset–Flare, AutoDock Vina, GOLD, and GLIDE were used for comparative analysis. Results highlighted the potential of drugs like leucovorin, nilotinib, ellence, thalomid, and carfilzomib against multiple cancer targets. Additionally, a library of novel molecules was constructed based on known drug scaffolds, with 20 prioritized based on drug-like properties. These molecules were further docked against specific cancer-related proteins, with AutoDock Vina yielding promising results [101].

Target-focused compound libraries are collections of compounds designed to interact with specific protein targets or related target families to identify potential drug candidates through screening. These libraries are crafted based on structural information about the target, chemogenomic models incorporating sequence and mutagenesis data, or knowledge of ligands interacting with the target. These libraries can serve as valuable resources for drug repurposing [103], for instance, Gan et al. introduced DrugRep, an automated and parameter-free tool designed for drug repurposing through receptor-based and ligand-based virtual screening [104]. Three drug libraries, consisting of approved drugs, experimental drugs, and traditional Chinese medicine, were collected for this purpose. DrugRep integrates innovative approaches such as CurPocket for receptor-based screening, which accurately predicts protein–ligand binding sites by calculating curvature factors. Additionally, LigMate and FitDock methods are incorporated for ligand-based virtual screening, offering improved enrichment power compared to other methods [104].

### 5.3. Transcriptome-Based Drug Repurposing, Using Gastrointestinal Stromal Tumors (GISTs) as an Example

Transcriptome-based drug repositioning has emerged as a promising strategy for identifying novel uses for existing drugs in cancer treatment. By analyzing gene expression profiles from cancer samples, researchers can identify drugs that have the potential to target specific pathways or molecular signatures associated with cancer [105,106]. One example is imatinib, a tyrosine kinase inhibitor initially developed to treat chronic myeloid leukemia (CML). Transcriptome analysis showed that GISTs often harbor mutations in the KIT receptor tyrosine kinase gene, leading to constitutive activation of KIT signaling. Imatinib was found to effectively inhibit KIT signaling, resulting in tumor regression in GIST patients, and was approved by the FDA [107,108,109].

### 5.4. Machine Learning (ML), Artificial Intelligence (AI), and Deep Learning (DL)

Machine learning (ML), artificial intelligence (AI), and deep learning (DL) leverage computational algorithms to analyze large datasets and identify patterns, enabling researchers to make predictions and decisions without explicit programming. ML algorithms are instrumental in the initial stages of drug discovery, particularly in identifying and validating potential therapeutic targets, compound screening and drug design, biomarker discovery, patient stratification, and clinical trial optimization. While these are not specifically focused on drug repurposing, the insights generated can inform the identification of existing drugs with potential repurposing opportunities for cancer treatment. ML, AI, and DL can be applied to literature searches, electronic health record (EHR)-based methods, and computational methods for drug-target interactions [110,111,112,113,114,115,116]. ML, AI, and DL are applied in various stages of drug discovery and development, including drug repurposing for cancer treatment (Figure 5).

Exploring PubMed abstracts with natural language processing (NLP) is a method utilized to uncover evidence of non-cancer drugs’ potential anticancer effects [105,106]. For instance, Zeng et al. developed RetriLite, a framework employing NLP and domain-specific knowledge to sift through papers and extract pertinent information [117]. This framework focuses on identifying papers discussing the efficacy of combination therapies in clinical or preclinical studies. In initial testing, RetriLite demonstrated high effectiveness with an F1 score of 0.93. Further validation centered on identifying agents enhancing antitumor efficacy with poly (ADP-ribose) polymerase inhibitors, where RetriLite achieved a 95.9% true positive rate and accurately distinguished between clinical and preclinical papers with 97.6% accuracy. Interobserver assessment confirmed the user consensus. RetriLite proved to be a valuable tool for establishing domain-specific information retrieval and extraction systems, equipping users with extensive metadata tags and keyword highlighting for efficient discovery in the combination therapy domain [117].

Datasets sourced from EHRs offer rich longitudinal and pathophysiological data, which significantly bolster efforts in drug repurposing [115,116]. One notable study by Ryu et al. focused on monoclonal gammopathy of undetermined significance (MGUS), which is a non-cancerous hematological condition that may develop into malignant diseases such as multiple myeloma [118]. Leveraging machine learning (ML) and EHR data, the researchers analyzed information from a comprehensive MGUS database encompassing 16,752 patients diagnosed between 2000 and 2021 at the Mayo Clinic. Integrating this data with medication and comorbidity information from the EHR, they scrutinized 21 drug classes of interest. Utilizing the XGBoost module, they trained a primary Cox survival model and performed sensitivity analyses on patient subsets. The analysis unveiled several medications linked to reduced risk of MGUS progression, including beta-blockers, immunosuppressants, multivitamins, non-coronary NSAIDS, opioids, proton pump inhibitors, statins, and vitamin D supplements [118].

Further, many novel computational techniques like time-resolved screening methods can be employed to study drug repurposing. Utilizing PubMed Identifier numbers (PMID), multiple time-resolved networks representing knowledge up to specific dates can be generated. These time-resolved networks can then be evaluated for computational repositioning by training on indications known during the network’s time period and testing on indications approved after that period, simulating real-world conditions more closely [119].

## 6. Future Direction

In the field of cancer treatment and drug repurposing, future directions involve finding new ways to improve how we treat cancer. This includes making treatments more personalized based on a person’s genes and molecules, strengthening immunotherapy against cancer, and discovering new targets for drugs to work better. Moreover, we will strive to combine different treatments, establish dependable markers to evaluate treatment efficacy, enhance drug delivery techniques, utilize computational analysis for improved data interpretation, and prioritize addressing the most pressing needs of patients. By working together globally and improving regulations, we hope to change how cancer is treated, aiming to help patients more and lessen the impact of cancer worldwide.

## 7. Conclusions

Drug repurposing is emerging as a promising avenue, offering advantages such as reduced development time and cost-effectiveness compared to novel drug discovery. By repositioning existing drugs for new clinical indications, researchers can leverage comprehensive safety and efficacy data, accelerating the transition to clinical practice. However, drug repurposing encounters challenges including patenting issues, financial constraints, and the necessity for combination therapies to overcome limitations of the monotherapy. Addressing the complexities and limitations of drug repurposing requires careful planning and collaboration across disciplines. By harnessing the collective expertise of researchers and leveraging cutting-edge technologies, we can unlock the full potential of drug repurposing in revolutionizing cancer therapeutics. Ultimately, these efforts offer renewed hope for patients and clinicians alike, paving the way for improved treatment outcomes and better quality of life.

## Figures and Tables

**Figure 1 cancers-16-01463-f001:**
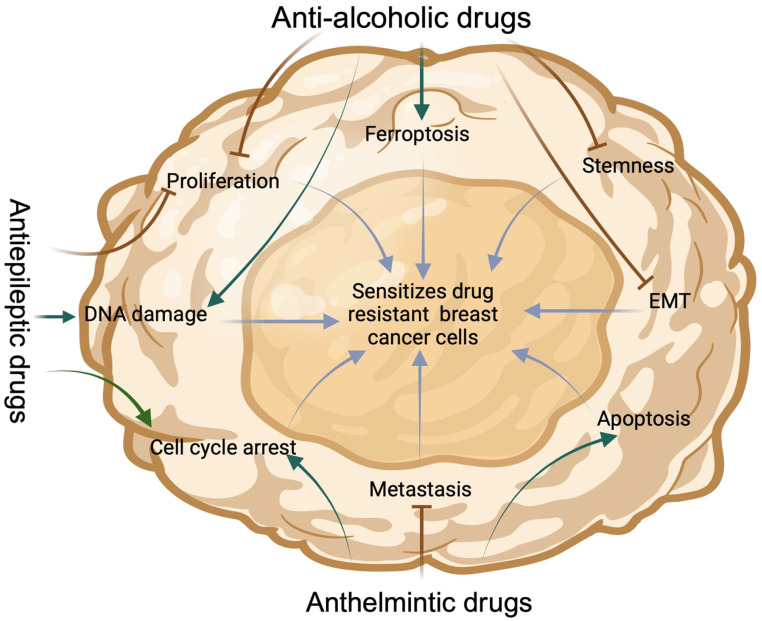
An illustration of the mechanism of action of anti-alcoholic drugs, antiepileptic drugs, and anthelmintic drugs in sensitizing breast cancer (BC) cells. Anti-alcoholic drugs sensitize BC cells by inducing DNA damage and ferroptosis, while also reducing proliferation, epithelial-to-mesenchymal transition (EMT), and stemness. Antiepileptic drugs sensitize BC cells by inducing DNA damage, causing cell cycle arrest, and inhibiting proliferation. Anthelmintic drugs sensitize BC cells by inducing cell cycle arrest and apoptosis, as well as inhibiting metastasis. Created with BioRender.com.

**Figure 2 cancers-16-01463-f002:**
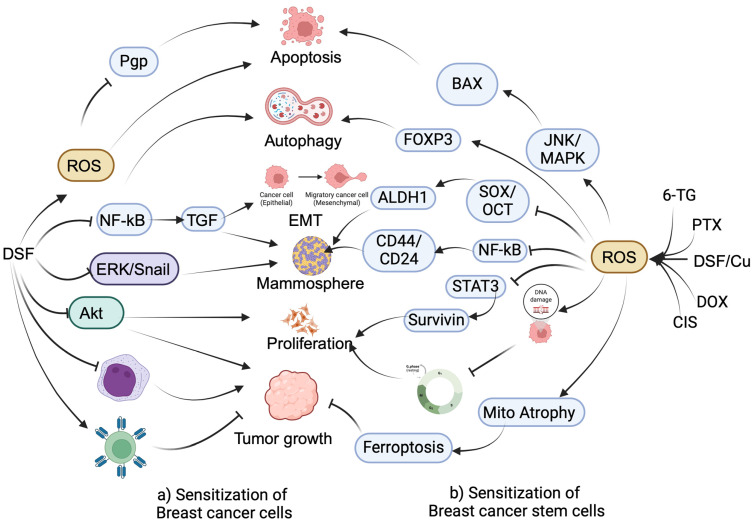
The combination of DSF/Cu and chemotherapeutic drugs sensitizes breast cancer (BC) cells and their stem cells by targeting various cellular and signaling mechanisms. (**a**) DSF sensitizes BC cells by inhibiting p-glycoprotein (Pgp) or inducing apoptosis and autophagy through the activation of reactive oxygen species (ROS). It also sensitizes BC cells by inhibiting epithelial-to-mesenchymal transition (EMT) and stemness via the targeting of NF-kB-dependent TGF. DSF further sensitizes BC cells by inhibiting ERK/Snail-dependent mammosphere formation and inhibiting BC cell proliferation by blocking the Akt pathway. Moreover, DSF enhances antitumor immunity by inhibiting myeloid-derived suppressor cells or activating CAR-T cells. (**b**) The combination of DSF/Cu with paclitaxel (PTX), doxorubicin (DOX), cisplatin (CIS), and 6-thioguanine (6-TG) triggers ROS, which in turn induces apoptosis by upregulating BAX expression via the JNK/MAPK signaling pathway in breast cancer stem cells (BCSCs). ROS also induces autophagy by activating FOXP3 signaling and reduces mammosphere formation in BCSCs by inhibiting ALDH1 via targeting transcription factors SOX2 and OCT4. Additionally, ROS inhibits the proliferation of BCSCs by inhibiting survivin via targeting STAT3 or inducing cell cycle arrest by causing DNA damage. Furthermore, ROS inhibits BCSCs within the tumor by inducing ferroptosis by triggering mitochondrial atrophy.

**Figure 3 cancers-16-01463-f003:**
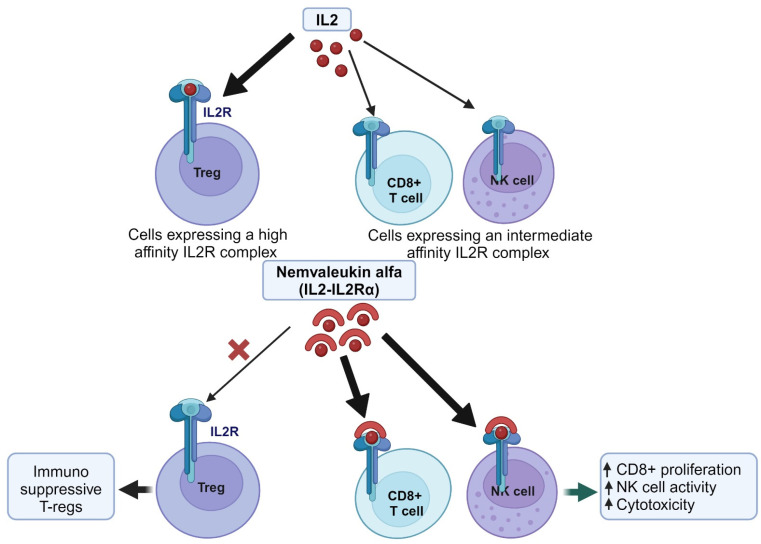
Example of re-engineering protein for drug repurposing. Nemvaleukin alfa, an IL-2-IL-2Rα fusion protein, binds to IL-2 receptor complexes. This leads to the activation of CD8-positive cytotoxic T cells and NK cells for increasing the cytotoxicity and decreasing the expansion of immunosuppressive Tregs. Created with BioRender.com.

**Figure 4 cancers-16-01463-f004:**
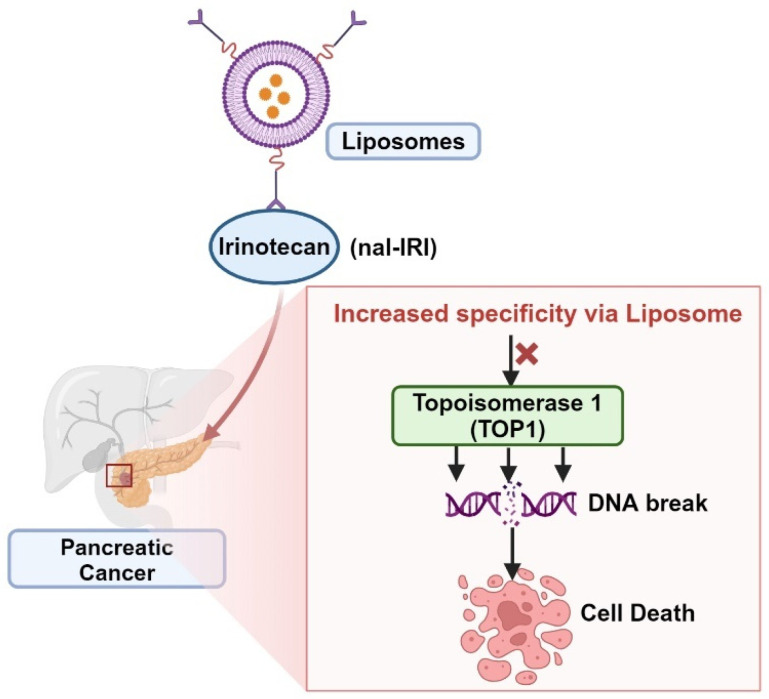
Increased specificity of nanoliposomal irinotecan (nal-IRI) induces DNA break in pancreatic cancer.

**Figure 5 cancers-16-01463-f005:**
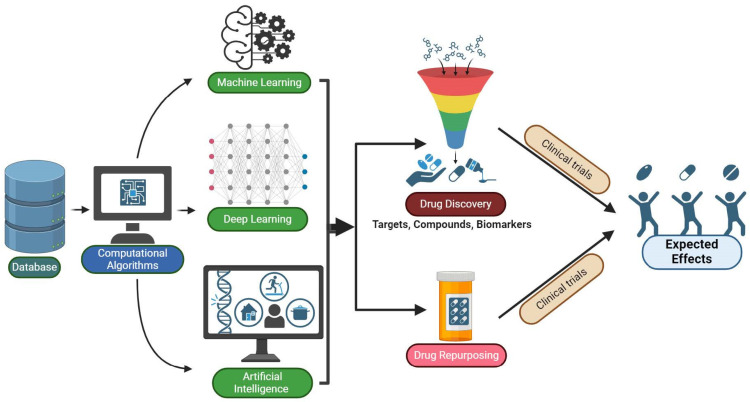
This schematic represents how machine learning (ML), artificial intelligence (AI), and deep learning (DL) are applied in various stages of drug discovery and development, including drug repurposing for cancer treatment. The diagram illustrates how computational algorithms analyze large datasets, leading to insights that inform different aspects of the drug discovery process, such as identifying therapeutic targets, compound screening, drug design, biomarker discovery, patient stratification, and clinical trial optimization. Created with BioRender.com.

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
