# Peer review of "Revitalizing Cancer Treatment: Exploring the Role of Drug Repurposing"

_cancers, 2024, doi:10.3390/cancers16081463_

Round 1

Reviewer 1 Report

Comments and Suggestions for Authors

In this work, the authors reported recent advances in drug repurposing, which covering experimental, re-engineering protein, nanotechnology and computational methods. The title of the manuscript is very general, but the content just summarized some specific examples, which should be adjusted. And some other comments also need to be addressed before publication.

1.       For the whole manuscript, it may be better to add more description for the repurposing method used in each example;

2.       For part of 2. Experimental approach, the author should include some advanced drug repurposing methods, such as time-resolved screening methods, which may be more interesting to the audience;

3.       There is a double space between (14). and increasingly for line 94, pls also check other similar errors throughout the manuscript;

4.       For the part of 4. Repurposing drugs with nanoliposomes, using pancreatic cancer as a case study, I don’t think it is repurposing strategy, it belongs to a delivery strategy.

5.       For part of 5.2. Structure based repurposing, Molsoft ICM is also one of widely used molecular modeling tool, Prof Chung-Hang Leung and his co-workers have done a lot of drug repurposing works using this computational technique, which should be included.

6.       The authors should use tables to compare different methods.

7.       Pls check the format of the references, especially journal name.

Comments on the Quality of English Language

Fine, some minor errors should be emitted.

Author Response

Dear Reviewer,

Thanks for your valuable suggestions.

Please find the comments below:

In this work, the authors reported recent advances in drug repurposing, which covering experimental, re-engineering protein, nanotechnology and computational methods. The title of the manuscript is very general, but the content just summarized some specific examples, which should be adjusted. And some other comments also need to be addressed before publication.

Comments: We thank the reviewer for insightful comments, please see our responses

  1. For the whole manuscript, it may be better to add more description for the repurposing method used in each example.

Comments: We thank the reviewer suggest, based on your suggestions to questions 2 and 5, we included descriptions. Further, we included section 5.3. Transcriptome-based drug repurposing, using Gastrointestinal Stromal Tumors (GIST) as an example.

  1. For part of 2. Experimental approach, the author should include some advanced drug repurposing methods, such as time-resolved screening methods, which may be more interesting to the audience;

Comments: Based on reviewer suggestion, we have added the following paragraph

“Further, many novel methods computational methods like time-resolved screening methods can be employed to study drug repurposing. Utilizing PubMed Identification numbers (PMID), multiple time-resolved networks representing knowledge up to specific dates can be generated. These time-resolved networks can then be evaluated for computational repositioning by training on indications known during the networks time period and testing on indications approved after that period, simulating real-world conditions more closely.”

  1. There is a double space between (14). and increasingly for line 94, pls also check other similar errors throughout the manuscript.

Comments: We thank the reviewer for catching this, we did our best to omit this.

  1. For the part of 4. Repurposing drugs with nanoliposomes, using pancreatic cancer as a case study, I don’t think it is repurposing strategy, it belongs to a delivery strategy.

Comments: We agree with reviewer that this is a delivery strategy, but this can be also considered as drug repurposing. Please see the following examples: https://doi.org/10.1002/VIW.20200127, https://www.nature.com/articles/s41392-020-00213-8, https://pubmed.ncbi.nlm.nih.gov/37939853/

  1. For part of 5.2. Structure based repurposing, Molsoft ICM is also one of widely used molecular modelling tool, Prof Chung-Hang Leung and his co-workers have done a lot of drug repurposing works using this computational technique, which should be included.

Comments: Based on reviewer suggestion, we have added the following paragraph

“Various commercial molecular docking tools such as CovDock, FLEXX, GOLD, ICM-Pro, DOCKTITE, Molecular operating environment (MOE), MacDOCK, CovalentDock, and AutoDock 4, are available for drug discovery and drug repurposing”.

  1. The authors should use tables to compare different methods.

Comments: We acknowledge reviewer suggestion, but this is beyond scope of this paper.

  1. Pls check the format of the references, especially journal name.

Comments: We thank the reviewer for pointing this, this is formatted by MDPI journal according their requirements, we did our best to use the updated citation via end note.

Reviewer 2 Report

Comments and Suggestions for Authors

Dear Authors,

You chose an extremely important subject in medicinal chemistry and not only, that of the discovery of new anticancer therapies, which might be lacked of the adverse reactions of the currently used drugs and of the resistance of cancer cells.

These therapies might be based on old drugs, used for different other pathologies, such as alcohol addiction, epilepsy or parasitic infections. You also presented in your review different other methodologies or technologies that might play a pivotal role in identifying potential candidates for repurposing.

In my opinion, the manuscript is well designed, well written, the figures are very clear and easy to follow and understand. I suggest reducing the size of Figures 1, 2 and 3.

The references are well chosen for the theme approached.

Author Response

Dear Reviewer,

Thanks for your suggestion and find the comments below:

  1. You chose an extremely important subject in medicinal chemistry and not only, that of the discovery of new anticancer therapies, which might be lacked of the adverse reactions of the currently used drugs and of the resistance of cancer cells.

Comments: Thanks for your positive comments.

  1. These therapies might be based on old drugs, used for different other pathologies, such as alcohol addiction, epilepsy or parasitic infections. You also presented in your review different other methodologies or technologies that might play a pivotal role in identifying potential candidates for repurposing.

Comments: Thanks for your positive comments.

  1. In my opinion, the manuscript is well designed, well written, the figures are very clear and easy to follow and understand. I suggest reducing the size of Figures 1, 2 and 3.

Comments: As per reviewer suggestion, we have reduced the size of figures (informed to editor).

  1. The references are well chosen for the theme approached.

Author comments: Thanks for your positive comments.

Reviewer 3 Report

Comments and Suggestions for Authors

Please justify your decision to discuss the anti-alcoholic, anthelmintic and antiepileptic drugs, but not others.

I suggest changing the title of the article to “Repurposing some (or several) anti-alcoholic, anthelmintic and antiepileptic drugs for cancer, using breast cancer as a case study”, because, obviously, not all anti-alcoholic, anthelmintic and antiepileptic drugs have anti-tumor activity, and the current title is too broad.

I assume that all figures were created using Biorender.com. Please mention it in all figure legends.

Line 100. “ identifying aspirin for TP53”. What does it mean??

Line 339. “Valproic acid (VPA) is a versatile antiepileptic drug”. I would say that VPA is, first of all, is HDAC inhibitor. Using HDAC inhibitors for cancer therapy is a well-established approach, such idea has been around for decades (https://pubmed.ncbi.nlm.nih.gov/18226465/, https://bmccancer.biomedcentral.com/articles/10.1186/s12885-016-2957-y, etc).

In other words, there are several classes of inhibitors that have been proposed for use in cancer therapy many years ago (lets say HDAC inhibitors, or others), and the mere fact that they can also be used as anti-alcoholic, anthelmintic and antiepileptic drugs does not make such “reposition” novel, as it's not repositioning for cancer, it's (at best) repositioning for breast cancer.

It seems like the authors choose an established anti-cancer drug and search if it can be used as anti-alcoholic, anthelmintic and antiepileptic drug. What's the point? The novelty might be their potential application for breast cancer although.

Speaking about another drug, Disulfiram, it inhibits alcoholdehydrogenase (hence can be used as anti-alcoholic drug), but it's role as anti-cancer drug is due to its copper-binding properties. Copper ionophores have been proposed to use as anti-cancer therapy ages ago (https://www.sciencedirect.com/science/article/abs/pii/S0006291X14004896), and Disulfiram action as a potential anti-breast cancer drug has been known for more than decade (https://journals.lww.com/eurjcancerprev/abstract/2014/05000/use_of_disulfiram_and_risk_of_cancer__a.10.aspx). The sub-chapter about anti-alcoholic drugs covers only two of them, Disulfiram and Naltrexone. Again, anti-cancer effects of Naltrexone have been known for long time.

And so on and so forth.

Thus, in my opinion, the review does not provide much novel information to the reader, also its title a bit misleading, as the authors mostly discuss repositioning of drugs (which are known anti-cancer drugs) for other types of cancer (admittedly, historically many of them were used - and still being used - for other diseases).

At the same time, the review lacks the vital and novel information about computational approaches to drug repositioning and novel tools for drug repositioning.

I suggest the authors elaborating on part of the text discussing AI for repositioning, computational tools, databases, etc. What about transcriptome-based drug repositioning?

Have the aforementioned computational tools identified Disulfiram and others as candidates for repositioning?

Author Response

Dear Reviewer,

Thanks for your valuable suggestions.

Please find the comments below. 

Reviewer 2 comments:

Please justify your decision to discuss the anti-alcoholic, anthelmintic and antiepileptic drugs, but not others.

  1. I suggest changing the title of the article to “Repurposing some (or several) anti-alcoholic, anthelmintic and antiepileptic drugs for cancer, using breast cancer as a case study”, because, obviously, not all anti-alcoholic, anthelmintic and antiepileptic drugs have anti-tumor activity, and the current title is too broad.

Comments: As per the suggestion of the reviewer, the correction was made in the page 3 and line 126.

  1. I assume that all figures were created using Biorender.com. Please mention it in all figure legends.

Comments: As per the suggestion of the reviewer, the correction was made in the legends of all figures.

  1. Line 100. “identifying aspirin for TP53”. What does it mean??

Comments:  As per the suggestion of the reviewer, the correction was made in the manuscript page 3 and line 107.

  1. Line 339. “Valproic acid (VPA) is a versatile antiepileptic drug”. I would say that VPA is, first of all, is HDAC inhibitor. Using HDAC inhibitors for cancer therapy is a well-established approach, such idea has been around for decades (https://pubmed.ncbi.nlm.nih.gov/18226465/, https://bmccancer.biomedcentral.com/articles/10.1186/s12885-016-2957-y, etc). In other words, there are several classes of inhibitors that have been proposed for use in cancer therapy many years ago (lets say HDAC inhibitors, or others), and the mere fact that they can also be used as anti-alcoholic, anthelmintic and antiepileptic drugs does not make such “reposition” novel, as it's not repositioning for cancer, it's (at best) repositioning for breast cancer. It seems like the authors choose an established anti-cancer drug and search if it can be used as anti-alcoholic, anthelmintic and antiepileptic drug. What's the point? The novelty might be their potential application for breast cancer although.

Comments:   As per the suggestion of the reviewer, the correction was made  and reference was included in the manuscript page 8 and line 349.

  1. Speaking about another drug, Disulfiram, it inhibits alcoholdehydrogenase (hence can be used as anti-alcoholic drug), but it's role as anti-cancer drug is due to its copper-binding properties. Copper ionophores have been proposed to use as anti-cancer therapy ages ago (https://www.sciencedirect.com/science/article/abs/pii/S0006291X14004896), and Disulfiram action as a potential anti-breast cancer drug has been known for more than decade (https://journals.lww.com/eurjcancerprev/abstract/2014/05000/use_of_disulfiram_and_risk_of_cancer__a.10.aspx). The sub-chapter about anti-alcoholic drugs covers only two of them, Disulfiram and Naltrexone. Again, anti-cancer effects of Naltrexone have been known for long time.

Comments: As per the suggestion of the reviewer, the correction was made  and reference was included in the manuscript page 4 and line 158 and 169.

  1. Thus, in my opinion, the review does not provide much novel information to the reader, also its title a bit misleading, as the authors mostly discuss repositioning of drugs (which are known anti-cancer drugs) for other types of cancer (admittedly, historically many of them were used - and still being used - for other diseases). At the same time, the review lacks the vital and novel information about computational approaches to drug repositioning and novel tools for drug repositioning. I suggest the authors elaborating on part of the text discussing AI for repositioning, computational tools, databases, etc. What about transcriptome-based drug repositioning? Have the aforementioned computational tools identified Disulfiram and others as candidates for repositioning?

Comments:

  • We have added the following paragraphs in computational approaches:

Section 5.2: Various commercial molecular docking tools such as CovDock, FLEXX, GOLD, ICM-Pro, DOCKTITE, Molecular operating environment (MOE), MacDOCK, CovalentDock, and AutoDock 4, are available for drug discovery and drug repurposing.

Section 5.4: Further, many novel methods computational methods like time-resolved screening methods can be employed to study drug repurposing. Utilizing PubMed Identification numbers (PMID), multiple time-resolved networks representing knowledge up to specific dates can be generated. These time-resolved networks can then be evaluated for computational repositioning by training on indications known during the networks time period and testing on indications approved after that period, simulating real-world conditions more closely”.

  • Based on reviewer suggestion, we have added a section on transcriptome-based drug repositioning

5.3. Transcriptome-based drug repurpsoing, using Gastrointestinal Stromal Tumors (GIST) as example.

Transcriptome-based drug repositioning has emerged as a promising strategy for identifying novel uses for existing drugs in cancer treatment. By analyzing gene expression profiles from cancer samples, researchers can identify drugs that have the potential to target specific pathways or molecular signatures associated with cancer105, 106. Imatinib, a tyrosine kinase inhibitor initially developed to treat chronic myeloid leukemia (CML). Transcriptome analysis showed that GISTs often harbor mutations in the KIT receptor tyrosine kinase gene, leading to constitutive activation of KIT signaling. Imatinib was found to effectively inhibit KIT signaling, resulting in tumor regression in GIST patients, and was approved by the FDA.

  • Disulfiram has been discussed in section 2.1.

Round 2

Reviewer 1 Report

Comments and Suggestions for Authors

Most of the comments are addressed. 

Author Response

Dear Reviewer,

I thank you for agreeing our comments and your valuable time for reviewing our article.

Kind Regards,

Nethaji M

Reviewer 3 Report

Comments and Suggestions for Authors

Line 605. "Further, many novel methods computational methods like time-resolved screening methods can be employed to study drug repurposing" - the authors use the word "methods" several times in the sentence. This is obviously a typographical error. Please fix it.

Author Response

Dear Reviewer,

I agree and thanks for your suggestion.

Line 605. "Further, many novel methods computational methods like time-resolved screening methods can be employed to study drug repurposing" - the authors use the word "methods" several times in the sentence. This is obviously a typographical error. Please fix it.

Comments:

Line 605 updated with track changes: Further, many novel computational techniques like time-resolved screening methods can be employed to study drug repurposing".

Thanks & Regards,

Nethaji M
